# Where do Reasoning Models make a Difference? Follow the Reasoning Leader for Efficient Decoding

## Abstract

Large reasoning models (LRMs) achieve strong reasoning performance by emitting long chains of thought. Yet, these verbose traces slow down inference and often drift into unnecessary detail, known as the overthinking phenomenon. To better understand LRMs' behavior, we systematically analyze the token-level misalignment between reasoning and non-reasoning models. While it is expected that their primary difference lies in the stylistic "thinking cues", LRMs uniquely exhibit two pivotal, previously under-explored phenomena: a ***Global Misalignment Rebound***, where their divergence from non-reasoning models persists or even grows as response length increases, and more critically, a ***Local Misalignment Diminish***, where the misalignment concentrates at the "thinking cues" each sentence starts with but rapidly declines in the remaining of the sentence. Motivated by the *Local Misalignment Diminish*, we propose **FoReaL-Decoding**, a collaborative fast-slow thinking decoding method for cost-quality trade-off. In FoReaL-Decoding, a Leading model leads the first few tokens for each sentence, and then a weaker draft model completes the following tokens to the end of each sentence. FoReaL-Decoding adopts a stochastic gate to smoothly interpolate between the small and the large model. On four popular math-reasoning benchmarks (AIME24, GPQA-Diamond, MATH500, AMC23), FoReaL-Decoding reduces theoretical FLOPs by 30 - 50% and trims CoT length by up to 40%, while preserving 86 - 100% of model performance. These results establish FoReaL-Decoding as a simple, plug-and-play route to controllable cost-quality trade-offs in reasoning-centric tasks.

## 1 Introduction

Reasoning has become a pivotal capability of large language models (LLMs), driving rapid progress in mathematical problem solving, code generation, and commonsense question answering (Huang & Chang, 2023; Ahn et al., 2024; Wang et al., 2024b; 2025b). Contemporary Large Reasoning Models (LRMs) such as OpenAI's GPT-o1 (OpenAI, 2024) and the open-source DeepSeek-R1 (DeepSeek-AI et al., 2025) demonstrate this trend by producing explicit long chains of thought (CoT) (Wei et al., 2023) that markedly improve performance on challenging tasks in mathematics (Xiong et al., 2025; Xia et al., 2025b), programming (Liu et al., 2024a), and other complex domains. These deeper, longer, and more precise reasoning trajectories represent the advanced "slow-thinking" patterns (Kahneman, 2011; Li et al., 2024d; 2025b). Although these slow-thinking LRMs showcase impressive reasoning skills, communities are increasingly concerned about the efficiency and fidelity of their often-lengthy chains of thought, a phenomenon known as "overthinking" (Chen et al., 2025c; Fan et al., 2025), where excessive computational resources are allocated for simple problems with minimal benefit.

To alleviate overthinking and improve efficiency, a series of methods has been proposed[1]. Most of these, however, require further post-training or manipulate the LRM's distribution itself, adding complexity or computational overhead. Motivated by Speculative Decoding (Leviathan et al., 2023) and the distinctions between fast and slow thinking, we ask: ***Where do reasoning models make a difference?*** To answer this, we first seek to pinpoint what truly differentiates strong reasoning models from standard instruction-following LLMs at the token level. For instruction-following models,

---

[1]The detailed related works are shown in Appendix C.

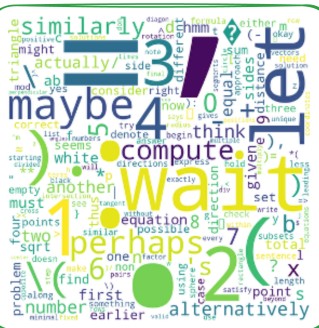

**Problem**
A regular hexagon can be divided into six equilateral triangles. If the perimeter of one of the triangles is 21 inches, what is the perimeter, in inches, of the regular hexagon?

**Response**
Okay, so I have this problem about a regular hexagon and some equilateral triangles. Let me try to understand what it's asking. It says a regular hexagon can be divided into six equilateral triangles. Hmm, I remember that a regular hexagon has all sides equal and all angles equal. If you divide it into six equilateral triangles, that must mean each triangle has sides equal to the sides of the hexagon. The problem gives me the perimeter of one of these triangles as 21 inches. I need to find the perimeter of the regular hexagon. Alright, let's break this down step by step. First, let me recall that the perimeter of a shape is the sum of all its sides. So, if each triangle is equilateral, all three sides are equal. That means each side of the triangle is the same length. If the perimeter is 21 inches, I can find the length of one side by …
Wait, let me make sure I got that right. If the hexagon is divided into six equilateral triangles, each triangle has sides equal to the sides of the hexagon. So, each triangle's side is the same as the hexagon's side. That makes sense because if you connect the center of the …

Figure 1: **Left**: An example comparing the token distribution alignment between *DeepSeek-R1-Distill-Qwen-32B* and *Qwen2.5-1.5B-Instruct*, qualitatively showing that the misaligned tokens (colored in red) are related to thinking patterns, and probably appear at the start of sentences. **Right**: The WordCloud of the misaligned tokens calculated on a mix of math datasets, quantitatively showing the high-frequency misaligned tokens like "*wait*", "*perhaps*", "*maybe*", "*let*", and "*alternatively*".

LIMA (Zhou et al., 2023) proposes the "superficial alignment" hypothesis, in which it shows that most of the knowledge has been learned in the pretraining and only a small amount of data is needed for alignment. Lin et al. (2023) further verifies this hypothesis from token-level analysis between the base model and the aligned model.

Leveraging the diagnostic framework of (Lin et al., 2023), our systematic analysis of misalignment across various model types (large reasoning, small reasoning, instruction-following, and pretrained model) reveals critical insights. We observe a "superficial alignment" phenomenon similar to (Lin et al., 2023), where misaligned tokens are predominantly stylistic (e.g., "*Wait*", "*Let me check*") rather than content-specific, often related to explicit thinking patterns. More strikingly, while previous work showed that misalignment between instruction-following and base models diminishes with longer context, we find this does not hold for reasoning models. Instead, we identify a ***Global Misalignment Rebound***, where overall misalignment between reasoning and non-reasoning models can slightly grow with response length, suggesting that increasing the length cannot reduce the misalignment. This indicates that the reasoning abilities are *not* as superficial as instruction-following. Crucially, despite this global trend, we uncover a corresponding ***Local Misalignment Diminish*** phenomenon: most token misalignments occur at the *beginning of each sentence*, then rapidly decrease until the next sentence starts. These findings reveal a novel *periodical, sentence-level misalignment diminishing pattern* unique to LRMs, driven by thinking-pattern indicators concentrated at sentence openings, shedding light on a better understanding of token-level divergences of these two types of models.

Based on this core insight that the reasoning pattern of LRMs is often front-loaded in each sentence, we hypothesize that strategic, limited intervention by a strong LRM can guide a weaker model, balancing reasoning quality with efficiency. To this end, we propose ***Follow the Reasoning Leader (FoReaL-Decoding)***, an efficient collaborative decoding method. In FoReaL-Decoding, a strong Leading model generates the initial few tokens of each sentence (capturing the potentially misaligned "thinking cues"), after which a weaker Draft model completes the sentence. To further mitigate potential overthinking from the Leading model (e.g., endlessly generating "Wait"), we introduce a stochastic binary gate that controls whether the Leading model intervenes. These two control knobs, lead token count and lead probability, allow FoReaL-Decoding to smoothly interpolate between the Draft and Leading models, offering strong controllability over the cost-quality spectrum.

**Contributions.** In summary, our primary contributions can be illustrated as follows:

- We conduct a systematic token-level analysis comparing LRMs with non-reasoning models, identifying two pivotal, under-explored phenomena: (1) ***Global Misalignment Rebound***, where the token distribution of LRMs diverges from that of non-reasoning models and their gap even increases with longer responses; (2) ***Local Misalignment Diminish***, where LRMs only make noticeable difference on generating stylistic "thinking-patterns" at the very beginning of each sentence. But such divergence from non-reasoning models rapidly drops on subsequent tokens within the sentence. This periodical *sentence-level misalignment diminishing* pattern has not been explored previously. These two discoveries significantly advance the understanding of LRMs.

- Leveraging these analytical insights (particularly the *Local Misalignment Diminish*), we propose *FoReaL-Decoding*, a training-free, collaborative algorithm that mixes the strength of a "slow-thinking" LRM (as Leading model) with the efficiency of a "fast-thinking", weaker model (as draft model). FoReaL-Decoding is designed to be plug-and-play, offering strong controllability to balance the cost and quality under diverse budgets of tokens.

- Experimental results on several reasoning-heavy math tasks (AIME24, GPQA-Diamond, MATH500, AMC23) demonstrate that FoReaL-Decoding reduces FLOPS by 30-55% and CoT length by up to 40%, while preserving 86-100% of the leading model's performance, effectively mitigating "overthinking".

## 2 TOKEN DISTRIBUTIONS OF REASONING VS. NON-REASONING MODELS

Large-scale reasoning models (LRMs) often outperform smaller instruction-tuned models on complex reasoning-heavy tasks, yet how their generation behavior differs from instruction models within the same model family remains unclear. (Lin et al., 2023) proposes an analytical method through the lens of token-distribution shifts and finds that alignments between instruction-following and base pretrained models are often superficial. This phenomenon is supported by nearly identical decoded tokens in the majority of token positions under the same input contexts, with distribution shifts occurring mainly with stylistic tokens like discourse markers. However, the critical question remains: "*Does this superficial alignment finding on instruction-following LLMs still hold for today's capable LRMs?*" Thus, our work systematically investigates token misalignment across various model combinations involving LRMs.

**Experimental Setup & Metric.** In this analysis, we utilize *DeepSeek-R1-Distill-Qwen-32B* as the targeting LRM, which we notate as the Leading model $P_L(\cdot)$. The corresponding small models, within the same family, that are used for comparison are noted as the Draft models $P_D(\cdot)$. The Draft models can be (i) the pretrained base model (*Qwen2.5-1.5B*), (ii) the instruction-following model (*Qwen2.5-1.5B-Instruct*), or (iii) the small reasoning model (*DeepSeek-R1-Distill-Qwen-1.5B*) in our analysis and method. For a user query $q$, the output response generated greedily from the Leading model can be notated as $y = \{y_1, ..., y_T\}$, where $T$ represents the length of the response. This response serves as the target for calculating the token distribution for the Draft model. At each position $t$, the context for predicting this token can be presented as $c_t = < q; y_{<t} >$, where $<;>$ represents the concatenation operation.

In the analysis, the aligned positions are defined as those token steps where the Draft model, when conditioned on the Leading model's history, would greedily generate exactly the same token as the Leading model, which means that *the two models have the same most probable behavior under the same context, indicating the alignment.*

Suppose $\mathcal{V}$ is the vocabulary for next-token prediction, then the aligned token indices are:

$$\mathcal{A} = \left\{ t \in \{1, \ldots, T\} \,:\, \arg\max_{y \in \mathcal{V}} P_D\big(y \mid c_t\big) = \arg\max_{y \in \mathcal{V}} P_L\big(y \mid c_t\big) \right\}, \tag{1}$$

which collects exactly those positions where the Draft model's top-1 prediction matches the Leading model's emitted token under the shared causal context $c_t$. Thus, the aligned and misaligned tokens can be defined:

$$\mathbf{y}_{\mathcal{A}} = \{ y_t \mid t \in \mathcal{A} \} \qquad \mathbf{y}_{\mathcal{A}^\complement} = \{ y_t \mid t \notin \mathcal{A} \} \tag{2}$$

**Qualitative Analysis on Misaligned Tokens.** Figure 1 (left) shows a qualitative example (truncated) from MATH500, comparing the token distribution alignment between *DeepSeek-R1-Distill-Qwen-32B* as the Leading model and *Qwen2.5-1.5B-Instruct* as the Draft model. The shown response $y$ is generated by the Leading model, the aligned tokens $\mathbf{y}_{\mathcal{A}}$ are colored in blue, and misaligned tokens $\mathbf{y}_{\mathcal{A}^\complement}$ are colored in red. Through the example, it can be intuitively perceived that the misaligned tokens are mostly stylistic tokens related to thinking patterns, and the beginning of each sentence[2] has a larger probability of being misaligned. To further quantitatively investigate what exactly these misaligned tokens are, we extract all the misaligned tokens from the mix of AIME24, AMC23, GPQA, and MATH datasets, count their frequencies, and generate the corresponding WordCloud shown in Figure 1 (right). From the WordCloud, it is observed that most of the high-frequency

---

[2]Sentences are defined as tokens separated by a period, question mark, exclamation mark, or newline symbol.

misaligned tokens are related to thinking patterns of LRMs, like "*wait*", "*perhaps*", "*maybe*", "*let*", and "*alternatively*", which shows a similar but different superficial phenomenon than previous instruction-following LLMs: While misalignment in both types of models is primarily stylistic rather than content-based, those in LRMs are distinctively characterized by tokens reflecting their overt reasoning or self-correction patterns. Thus, our qualitative exploration reveals that LRM misalignment is characterized by stylistic "thinking cues" concentrated at sentence beginnings, prompting a more detailed quantitative analysis of their underlying distribution patterns.

**Global Misalignment Rebound.** Existing analysis on token distribution shifts between instruct and base models has identified that such shifts will gradually diminish over time during the decoding process due to the more comprehensive context given, as shown in Figure 2 (upper, red line). In the figure, the y-axis represents the average misalignment rate at each token position, while the x-axis represents the token position within the whole response (upper panel) or sentence (lower panel). As shown, the red line, representing misalignment between the instruct model and base model, decreases and remains at a low rate. This implies that providing longer context can gradually compensate for the misalignment between instruct and base models.

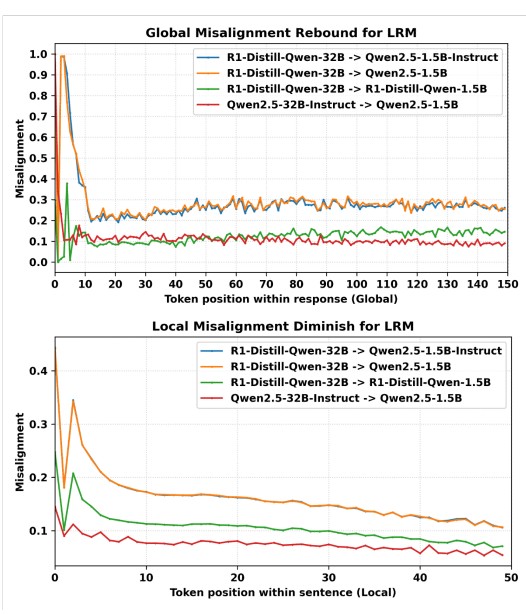

However, this response-level misalignment diminishing phenomenon does not strictly hold for LRMs. As illustrated in Figure 2 (upper), lines corresponding to LRM as the Leading model exhibit different behaviors. When the Draft models are instruct (blue line) or base (orange line) models, the misalignment rates initially decrease dramatically to around $0.2$, then rebound and persist around $0.3$. In contrast, the green line, representing misalignment between large and small reasoning models (which belong to the same family and are trained on similar data), shows consistently low misalignment from the beginning, indicating a distinct trend. We term the observed persistent or rebounding divergence between LRMs and non-reasoning models the ***Global Misalignment Rebound*** phenomenon. This phenomenon, characteristic of LRM comparisons with non-reasoning models, is mainly

Figure 2: **Top**: Response-level misalignment changes with response length. **Bottom**: sentence-level misalignment changes with response length. The y-axis represents the average misalignment rate at each token position, the x-axis represents the token position within the whole response or sentence. We reveal the novel Global Misalignment Rebound and Local Misalignment Diminish phenomenon that only occurs on current LRMs, shown as the blue, orange, and green lines of the upper figure. This phenomenon does not hold for the previous alignment between the instruction-following and base models, shown in the red line.

caused by LRMs continuously generating thinking patterns at the beginning of sentences, partly to prevent premature conclusion of the generation process. This finding demonstrates that merely extending context length is insufficient to resolve the misalignment between reasoning and non-reasoning models, indicating that reasoning capability is *not* as superficial as instruction-following.

**Local Misalignment Diminish.** It is uncommon that a longer context does not benefit the alignment. Thus, to further understand this behavior, we conduct the sentence-level analysis by calculating the token misalignment rate at each sentence-level position. In the response, sentences can be separated by periods, question marks, exclamation marks, and the newline symbol. Specifically, for any position $x$, we first collect every sentence that is at least $x$ tokens long. Mark the $x$-th token in each of those sentences as $1$ if it is misaligned and $0$ if it is aligned. The average of these 0-1 indicators across all selected sentences is the misalignment rate for position $x$.

As shown in Figure 1 (lower), for the red line, there is no obvious misalignment decrease that can be observed. It means that between the instruct and the base model, the misalignment occurs relatively evenly across the whole sentence. On the contrary, for LRM-involved model combinations, the blue, orange, and green lines, the misalignment rates drop dramatically at the first several tokens,

e.g., from $0.4$ to $0.15$, and then keep diminishing, indicating a totally different behavior. Thus, we term this phenomenon the ***Local Misalignment Diminish*** phenomenon for reasoning models. These findings reveal a novel periodical, sentence-level misalignment diminish pattern unique to LRMs, driven by thinking-pattern indicators concentrated at sentence openings, shedding light on a better understanding of token-level divergences of these two types of models.

**Findings.** From this section, several key findings can be concluded:

- LRM misalignment with non-reasoning models, while largely superficial and characterized by stylistic "thinking cues", uniquely exhibits a ***Global Misalignment Rebound***. Unlike instruct models that increasingly align with more context, token divergence at the response level can persist or even grow, underscoring deeper, ingrained differences in their generative behavior.

- LRMs distinctively display a ***Local Misalignment Diminish***. This manifests as a novel, periodical sentence-level pattern where high misalignment, driven by "thinking cues" concentrated at sentence beginnings, rapidly decreases as the sentence progresses. This predictable intra-sentence dynamic is a crucial insight for developing LRM-guided decoding and understanding LRM patterns.

## 3 FoReaL-Decoding

Motivated by the above token divergence analysis, we propose a collaborative fast-slow thinking decoding method for cost-quality Trade-off, ***Follow the Reasoning Leader (FoReaL-Decoding)***, a plug-and-play training-free method that mixes the strength of a slow but highly capable large reasoning model with the speed of a small model. The central idea is to let the strong, large (*Leading*) model lead at the beginning of sentences, and allow the weaker, small (*Draft*) model to complete the rest of the tokens. This decoding algorithm is of strong controllability, which can smoothly transfer into the Leading model only or downgrade to the Draft model only, by controlling the probability and the number of tokens to lead[3].

**Preliminaries.** The two control knobs that govern the trade-off between cost and quality: *Required lead count* $n \in \mathbb{N}$: the minimum number of tokens the Leading model generates before yielding control to the Draft model. *Lead probability* $p \in [0, 1]$: probability that a sentence is led by the Leading model. When $p = 0$, the decoding system degenerates to pure Draft model decoding; when $p = 1$ and $n$ exceeds the sentence length, it transfers to Leading model decoding. Intermediate settings form a *continuity* of compute–accuracy trade-offs.

In addition, let $t \in \mathbb{N}$ represent the global token index in the response, and $s \in \mathbb{N}$ represent the sentence index. $g_s \sim \text{Bernoulli}(p)$ represents the sentence-level gate to decide what model to start the sentence $s$: the sentence will be led by the Leading model if $g_s = 1$. $\tau_s$ represents the global position of the first token in $s$. $s(t) = \max\{s : \tau_s \leq t\}$ is the function that maps the token $t$ to the sentence index that $t$ belongs to. $\lambda_t = t - \tau_{s(t)} + 1$ is the local position of token $t$ within its sentence.

**Intra-Sentence Lead** Within a sentence $s$, the generation of each token at position $t$ is governed by the token-level policy,

$$
\pi_t = \begin{cases} L & g_{s(t)} = 1 \ \wedge \ \left[\lambda_t \leq n \ \vee \ t < H_{s(t)}^{\text{hit}}\right], \\ D & \text{otherwise.} \end{cases} \tag{3}
$$

$g_{s(t)} = 1$ represents this sentence $s(t)$ should be led by the Leading model, decided by the gate. $L$ and $D$ represent the Leading model and Draft model, respectively. $\lambda_t \leq n$ represents the index of this token within this sentence that is smaller than the required lead count $n$, thus should be generated by the Leading model. $H_s^{\text{hit}}$ is the first token index within $s$ where the top-1 token generated by the Draft model matches that of the Leading model for $k$ consecutive steps:

$$
H_s^{\text{hit}} = \min\{t : s(t) = s, \lambda_t > n, h_t = k\}, \tag{4}
$$

where $h_t$ represents the number of consecutive hits within the max sliding window of $k$:

$$
h_t = \sum_{i=0}^{k-1} \delta_{t-i}, \quad \delta_t = \mathbf{1}\left\{\underset{y \in \mathcal{V}}{\arg\max} P_D(\cdot|c_t) = \underset{y \in \mathcal{V}}{\arg\max} P_L(\cdot|c_t)\right\} \tag{5}
$$

---

[3]The detailed pseudo code is shown in Appendix A.

Put it simply, for each sentence, if the Bernoulli gate decides to let $P_L$ lead the sentence with the probability $p$, $P_L$ will generate the first $n$ tokens. Then, $P_D$ begins the generation process as well, with the purpose of measuring the alignment between the two models. When the top-1 predictions of these two models aligned with each other for $k$ times, the generation process is yielded to $P_D$, otherwise, $P_L$ generates the whole sentence. On the contrary, if the gate decides not to let $P_L$ lead, then the whole sentence will be completely generated by $P_D$.

**Sentence-level likelihood.**  For sentence $s$ with token span $Y_s = (y_{\tau_s}, \ldots, y_{\tau_{s+1}-1})$ and length $L_s$, the conditional likelihood under FoReaL-Decoding is:

$$P_{\text{CoL}}(Y_s \mid g_s) = \prod_{i=0}^{L_s-1} P_{\pi_{\tau_s+i}}(y_{\tau_s+i} \mid c_{\tau_s+i}), \qquad (6)$$

Whenever $\pi_t = L$, the factor draws its probability from the distribution $P_L$ of the Leader model; otherwise from the Draft model of distribution $P_D$.

**Inter-Sentence Transfer**  Upon encountering a sentence boundary at the token $t$, i.e., the sentence is complete, we execute the inter-sentence update by resetting the hit counter and resampling the gate for the next sentence.

$$s \leftarrow s+1, \quad g_s \sim \text{Bernoulli}(p), \quad h_t \leftarrow 0 \qquad (7)$$

## 4 EXPERIMENTS

### 4.1 IMPLEMENTATION DETAILS

**Models, Datasets, and Setup.**  To assess the effects of FoReaL-Decoding, extensive experiments are conducted for different model combinations in the Qwen2.5 family, including reasoning models like *R1-Distill-Qwen-32B* (DeepSeek-AI et al., 2025), *R1-Distill-Qwen-1.5B* (DeepSeek-AI et al., 2025), non-reasoning instruct models like *Qwen2.5-7B-Instruct* (Team, 2024), *Qwen2.5-1.5B-Instruct* (Team, 2024), and base models like *Qwen2.5-1.5B* (Team, 2024). To cover a wide scope of potential trade-offs, we utilize the reasoning models as the Leading models, while any of the above types as the Draft models. Moreover, our extensive experiments on the recently released *Qwen3* (Team, 2025) series further verify the generalizability of our method. We evaluate our method on relatively hard, reasoning-heavy math datasets, including AIME2024 (AI-MO, 2024a), GPQA-Diamond (Rein et al., 2024), AMC23 (AI-MO, 2024b), and MATH500 (Lightman et al., 2023). All experiments were conducted on NVIDIA A100 GPUs (80G), utilizing the Huggingface Transformers package. During the generation, we follow the recommended generation configuration from R1-Distill models as `temperature=0.6`, `top_p=0.95`, `top_k=40` for all the experiments. During the generation, we always let the Leading model generate the first paragraph, and we fix the required hits for generation transfer as $k = 5$ for all the experiments.

### 4.2 MAIN RESULTS

Table 1 presents the comparisons between accuracy and efficiency (TFLOPs) of FoReaL-Decoding on commonly used reasoning-heavy math problem tasks. We provide some different configurations as controls to show the wide trade-off scopes of our method. We also present the reported results of the concurrent work, Speculative Thinking (Yang et al., 2025), for better comparison. The accuracies on each line are compared with the Draft model, and the TFLOPs are compared with the Leading models: better values are colored in green, otherwise red. We utilize the theoretically estimated TFLOPs[4] as the efficiency measurement since it takes the generation length into account, different from the estimated speed. In the main comparison, we focus on 3 collaborative settings. Across four benchmarks, FoReaL-Decoding cuts inference cost by $30 - 55\%$ relative to Leader-only decoding while retaining 86 - 100% of its accuracy. The detailed statistics, including response length and leading ratios on AIME24, can be found in Table 2 for better understanding.

*R1-Distill-Qwen-32B for Leading, R1-Distill-Qwen-1.5B for Draft.* This collaborative setting yields the highest accuracies for all of the math reasoning datasets. In this setting, the larger 32B reasoning model takes charge of the leading of the sentences, while the smaller 1.5B reasoning model needs to

---

[4]The estimated TFLOPs are calculated based on the detailed formulas presented in Appendix B.

Table 1: Comparisons of Accuracy and Efficiency (TFLOPs) of FoReaL-Decoding on commonly used reasoning-heavy math problem tasks. To further show the wide trade-off scopes of our method, we provide some different configurations as the control. The results of Speculative Thinking are the reported results. The accuracies are better with higher (↑) values, while the TFLOPs are better with lower (↓) values. The accuracies on each line are compared with the Draft model, and the TFLOPs are compared with the Leading models: better values are colored in green, otherwise red.

| Model | | AIME24 | | GPQA-D | | MATH500 | | AMC23 | |
|---|---|---|---|---|---|---|---|---|---|
| Method | Config | ACC (%) ↑ | TFLOPs ↓ | ACC (%) ↑ | TFLOPs ↓ | ACC (%) ↑ | TFLOPs ↓ | ACC (%) ↑ | TFLOPs ↓ |
| **DeepSeek-R1-Distill-Qwen-32B + DeepSeek-R1-Distill-Qwen-1.5B** | | | | | | | | | |
| DeepSeek-R1-Distill-Qwen-32B | | 66.7 | 15.72 | 59.6 | 8.09 | 93.6 | 4.13 | 95.0 | 7.54 |
| DeepSeek-R1-Distill-Qwen-1.5B | | 23.3 | 2.86 | 22.2 | 1.13 | 81.4 | 1.14 | 65.0 | 2.51 |
| Speculative Thinking | | 32.2 | 41.9 | 41.9 | 2.62 | 89.4 | 1.51 | 80.0 | 3.31 |
| FoReaL-Decoding | $n{=}15,p{=}0.4$ | 33.3 (+10.0) | **5.60** (-10.12) | 43.3 (+21.1) | **2.47** (-5.62) | 90.2 (+8.8) | **1.43** (-2.88) | 80.0 (+15.0) | **2.91** (-4.63) |
| FoReaL-Decoding | $n{=}15,p{=}0.6$ | 50.0 (+26.7) | 6.77 (-8.95) | 48.2 (+26.0) | 4.50 (-3.59) | 91.4 (+10.0) | 2.40 (-1.26) | 80.0 (+15.0) | 3.99 (-3.55) |
| FoReaL-Decoding | $n{=}15,p{=}0.8$ | 50.0 (+26.7) | 8.47 (-7.25) | 54.6 (+32.4) | 4.69 (-3.40) | 93.4 (+12.0) | 2.70 (-1.43) | 90.0 (+25.0) | 5.37 (-2.17) |
| FoReaL-Decoding | $n{=}15,p{=}1.0$ | **66.7** (+43.4) | 9.16 (-6.56) | 56.6 (+34.4) | 6.21 (-1.88) | 93.2 (+11.8) | 3.14 (-0.99) | 92.5 (+27.5) | 5.28 (-2.26) |
| FoReaL-Decoding | $n{=}25,p{=}0.8$ | 53.3 (+30.0) | 10.95 (-4.77) | 57.1 (+34.9) | 5.65 (-2.44) | 92.6 (+11.2) | 3.13 (-1.00) | 92.5 (+27.5) | 4.99 (-2.55) |
| FoReaL-Decoding | $n{=}25,p{=}1.0$ | **66.7** (+43.4) | 10.54 (-5.18) | **57.6** (+35.4) | 6.68 (-1.41) | **94.5** (+13.1) | 3.50 (-0.63) | **95.0** (+30.0) | 5.66 (-1.88) |
| **DeepSeek-R1-Distill-Qwen-32B + Qwen2.5-1.5B-Instruct** | | | | | | | | | |
| DeepSeek-R1-Distill-Qwen-32B | | 66.7 | 15.72 | 59.6 | 8.09 | 93.6 | 4.13 | 95.0 | 7.54 |
| Qwen2.5-1.5B-Instruct | | 0.0 | 0.12 | 23.7 | 0.12 | 49.2 | 0.09 | 15.0 | 0.10 |
| FoReaL-Decoding | $n{=}15,p{=}0.8$ | 20.0 (+20.0) | **9.05** (-6.67) | 38.4 (+14.7) | 5.63 (-2.46) | 76.2 (+27.0) | 2.85 (-1.28) | 65.0 (+50.0) | 5.22 (-2.32) |
| FoReaL-Decoding | $n{=}15,p{=}1.0$ | 20.0 (+20.0) | 11.19 (-4.53) | 47.5 (+23.8) | 5.86 (-2.23) | 85.9 (+36.7) | 3.28 (-0.85) | 85.0 (-70.0) | 6.15 (-1.39) |
| FoReaL-Decoding | $n{=}25,p{=}0.8$ | 36.7 (+36.7) | 9.58 (-6.14) | 45.0 (+21.3) | **4.37** (-3.72) | 82.0 (+32.8) | **2.52** (-1.61) | 72.5 (+57.5) | **4.65** (-2.89) |
| FoReaL-Decoding | $n{=}25,p{=}1.0$ | **40.0** (+40.0) | 11.00 (-4.72) | **57.1** (+33.4) | 6.27 (-1.82) | **90.8** (+2.8) | 3.36 (-0.77) | 92.5 (-77.5) | 6.88 (-0.66) |
| **DeepSeek-R1-Distill-Qwen-1.5B + Qwen2.5-7B-Instruct** | | | | | | | | | |
| DeepSeek-R1-Distill-Qwen-1.5B | | 23.3 | 2.86 | 22.2 | 1.13 | 81.4 | 1.14 | 65.0 | 2.51 |
| Qwen2.5-7B-Instruct | | 6.7 | 0.95 | 38.4 | 0.89 | 76.0 | 0.61 | 52.5 | 0.75 |
| Speculative Thinking | | 6.7 | 4.93 | 31.8 | 6.73 | 74.8 | 2.04 | 55.0 | 4.97 |
| FoReaL-Decoding | $n{=}15,p{=}0.8$ | 16.7 (+10.0) | 2.05 (-0.81) | **34.3** (-4.1) | 1.07 (-0.06) | 76.4 (+0.4) | 0.57 (-0.57) | 57.5 (+5.0) | **1.08** (-1.43) |
| FoReaL-Decoding | $n{=}15,p{=}1.0$ | 16.7 (+10.0) | 6.47 (+3.61) | 29.8 (-8.6) | 3.08 (+1.95) | **79.6** (+3.6) | 1.42 (+0.28) | 52.5 (+0.0) | 3.35 (+0.84) |
| FoReaL-Decoding | $n{=}25,p{=}0.8$ | 20.0 (+13.3) | **1.57** (-1.29) | 33.3 (-5.1) | **0.80** (-0.33) | 78.6 (+2.6) | **0.55** (-0.59) | 65.0 (+12.5) | 1.76 (-0.75) |
| FoReaL-Decoding | $n{=}25,p{=}1.0$ | **23.3** (+16.6) | 3.18 (+0.32) | 29.3 (-9.1) | 2.53 (+1.40) | 79.2 (+3.2) | 1.04 (-0.10) | 65.0 (+12.5) | 1.66 (-0.85) |

complete the remaining sentence. In this setting, both models have the reasoning capabilities, but FoReaL-Decoding implicitly separates the generation of each sentence into two phases and yields the less informative Draft phase to the smaller model for better efficiency. As shown in the table, all our results obtain better performances compared with the Draft model and efficiencies compared with the Leading model, and also exceed Speculative Thinking, indicating the capability of our methods. Moreover, on all of the tasks except GPQA-D, FoReaL-Decoding reaches similar or even slightly higher performances than the 32B Leading model with fewer TFLOPs.

*R1-Distill-Qwen-32B for Leading, Qwen2.5-1.5B-Instruct for Draft.* This setting represents a direct mixture of a large reasoning model and a small non-reasoning model. As shown in the table, the 1.5B instruct model performs badly on the given difficult math problems. The use of a stronger reasoning model for leading largely improves the accuracy, although with more computation required. The response lengths are largely shorter than *R1-Distill-Qwen-1.5B*, representing an alleviation of overthinking. Compared with using another small reasoning model for Draft, utilizing the instruction model leads to suboptimal performance. To understand this phenomenon, further experiments are conducted where the base pretrained model *Qwen2.5-1.5B* is utilized as the Draft model. As shown in Table 2, the accuracies, response lengths, and estimated TFLOPs are almost identical compared with

Table 2: The detailed results of different collaborative settings on AIME. Additional configuration that uses base model for Draft is included.

| Model | | AIME24 | | | |
|---|---|---|---|---|---|
| Method | Config | ACC (%) | Length | Ratio | TFLOPs |
| **DeepSeek-R1-Distill-Qwen-32B + DeepSeek-R1-Distill-Qwen-1.5B** | | | | | |
| FoReaL-Decoding | $n{=}15,\ p{=}0.4$ | 33.3 | 11 876 | 0.272 | 5.60 |
| FoReaL-Decoding | $n{=}15,\ p{=}0.6$ | 50.0 | 10 934 | 0.401 | 6.77 |
| FoReaL-Decoding | $n{=}15,\ p{=}0.8$ | 50.0 | 11 532 | 0.527 | 8.47 |
| FoReaL-Decoding | $n{=}15,\ p{=}1.0$ | 66.7 | 10 617 | 0.666 | 9.16 |
| FoReaL-Decoding | $n{=}25,\ p{=}0.8$ | 53.3 | 12 081 | 0.676 | 10.95 |
| FoReaL-Decoding | $n{=}25,\ p{=}1.0$ | 66.7 | 11 116 | 0.683 | 10.54 |
| **DeepSeek-R1-Distill-Qwen-32B + Qwen2.5-1.5B-Instruct** | | | | | |
| FoReaL-Decoding | $n{=}15,\ p{=}0.8$ | 20.0 | 12 584 | 0.571 | 9.05 |
| FoReaL-Decoding | $n{=}15,\ p{=}1.0$ | 20.0 | 14 188 | 0.588 | 11.19 |
| FoReaL-Decoding | $n{=}25,\ p{=}0.8$ | 36.7 | 11 575 | 0.710 | 9.58 |
| FoReaL-Decoding | $n{=}25,\ p{=}1.0$ | 40.0 | 11 239 | 0.813 | 11.00 |
| **DeepSeek-R1-Distill-Qwen-32B + Qwen2.5-1.5B (Base)** | | | | | |
| FoReaL-Decoding | $n{=}15,\ p{=}0.8$ | 23.3 | 12 224 | 0.547 | 9.56 |
| FoReaL-Decoding | $n{=}15,\ p{=}1.0$ | 20.0 | 12 107 | 0.664 | 10.39 |
| **DeepSeek-R1-Distill-Qwen-1.5B + Qwen2.5-7B-Instruct** | | | | | |
| FoReaL-Decoding | $n{=}15,\ p{=}0.8$ | 16.7 | 4 120 | 0.545 | 2.05 |
| FoReaL-Decoding | $n{=}15,\ p{=}1.0$ | 16.7 | 14 132 | 0.651 | 6.47 |
| FoReaL-Decoding | $n{=}25,\ p{=}0.8$ | 20.0 | 4 474 | 0.693 | 1.57 |
| FoReaL-Decoding | $n{=}25,\ p{=}1.0$ | 23.3 | 11 436 | 0.841 | 3.18 |

using base and instruct models, which means the previous instruction-aligned process does not benefit the current reasoning settings. Moreover, we also include the results of using Qwen3 family models with similar settings, which shows much better performance due to the stronger capabilities.

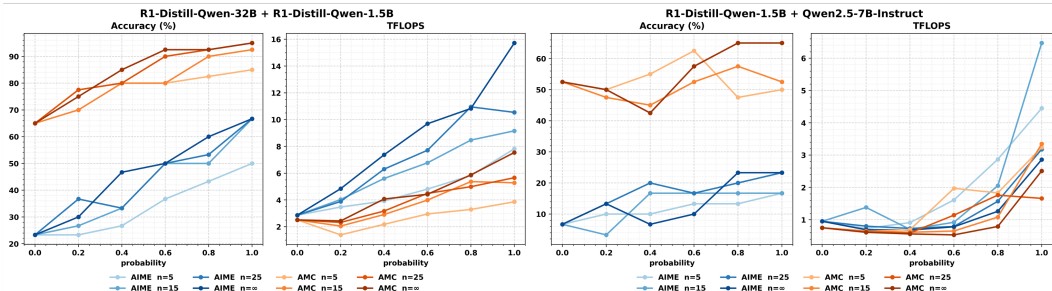

Figure 3: Effects of lead count and lead probability on AIME24 and AMC23 datasets, based on 2 collaborative configurations. FoReaL-Decoding provides a smooth cost-quality trade-off, making the transition from the weak Draft model to the strong Leading model smooth and controllable.

*R1-Distill-Qwen-1.5B for Leading, Qwen2.5-7B-Instruct for Draft.* Different from the above settings, in which a strong but large reasoning model is used as the Leading model, this setting considers a different and most efficient scenario, utilizing a small reasoning model for leading and a slightly larger instruct model for Draft. In this setting, the efficiencies are reduced to an extremely low level, even faster than directly utilizing the small reasoning models. As shown in Table 2, FoReaL-Decoding largely reduces the length required for the problem, thus largely reducing the computation required. On AIME24 and AMC23, our method reaches the same accuracy as the Leading model with similar or less computation. On GPQA, our method reaches an intermediate accuracy, since the abnormal situation where a non-reasoning model has better performance than the reasoning model.

## 4.3 EFFECTS OF LEAD COUNT AND LEAD PROBABILITY

Figure 3 sweeps the two hyperparameters that govern the controllability of FoReaL-Decoding, lead count $n$ and lead probability $p$ on AIME24 and AMC23 datasets, based on 2 collaborative configurations, *DeepSeek-R1-Distill-Qwen-32B + DeepSeek-R1-Distill-Qwen-1.5B* and *DeepSeek-R1-Distill-Qwen-1.5B + Qwen2.5-7B-Instruct*, representing the high-performance and high-efficiency settings, respectively. For each model combination, we run experiments on $n \in \{5, 15, 25, +\infty\}, p \in \{0, 0.2, 0.4, 0.6, 0.8, 1.0\}$. When $p = 0$, FoReaL-Decoding utilizes the Draft model only, and utilizes the Leading model only when $p = 1$ and $n = +\infty$. According to the figure, FoReaL-Decoding provides a smooth cost-quality trade-off, making the transition from the weak Draft model to the strong Leading model smooth and controllable. For any fixed $n$, increasing the probability $p$ of the Leader intervention shifts the operating point up and to the right: accuracy rises while estimated TFLOPs grows almost linearly. The resulting curve is smooth, allowing practitioners to trade latency for quality by adjusting $(n, p)$. The jump from $n = 5$ to $n = 15$ yields large accuracy gains at a modest cost increase. Further enlarging the Leader count ($n \geq 25$) adds little accuracy yet inflates compute up a lot, indicating that sentence-level guidance already captures most of the benefit of slow reasoning.

## 4.4 TRADE-OFF CURVES

Figure 4 plots the trade-off curves between accuracy and estimated TFLOPs for every $(n, p)$ configuration tested on AIME24 (left) and AMC23 (right), according to our experiment scopes on Qwen2.5 family. Blue markers correspond to our variants, red circles denote the corresponding LRMs, and the dashed line is the empirically computed Pareto frontier. On both benchmarks, every LRM point is Pareto dominated: an alternative FoReaL-Decoding setting always achieves higher accuracy at lower cost. Moreover, we find that the frontier rises sharply between $0.5$ and $2$ estimated TFLOPs, as each additional estimated TFLOPs yields 10-15 percentage points of accuracy. However, beyond

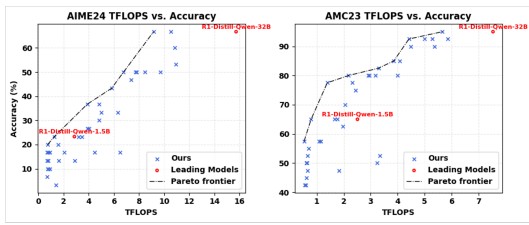

Figure 4: The trade-off curves between accuracy and estimated TFLOPs. Blue markers correspond to our variants, red circles denote the corresponding LRMs, and the dashed line is the empirically computed Pareto frontier. On both benchmarks, every LRM point is Pareto dominated.

$\approx 5$ estimated TFLOPs, the curve flattens; extra compute buys only marginal improvements up to the ceiling.

## 4.5 Results on Qwen3 Families

To further verify the effectiveness and generalizability of FoReaL-Decoding on otehr models, additional experiments are conducted on the Qwen3 series of models, including *Qwen3-32B*, *Qwen3-1.7B*, and *Qwen3-0.6B*, due to the various sizes of models provided in the family. Since Qwen3 family models have both reasoning and non-reasoning modes, we utilize both modes for these models and follow exactly the same generation configuration as our main experiments. The detailed experimental results are shown in Table 3. As shown in the table, FoReaL-Decoding shows promising performances on all the configurations. Most notably, in the *Qwen3-32B (reasoning) + Qwen3-0.6B (non-reasoning)* configuration, FoReaL-Decoding reaches a similar accuracy (66.6% to 73.3%) with only half of the estimated TFLOPs and with less response length. Compared with the utilzing Qwen2.5 family models using reasoning plus non-reasoning models, Qwen3 family models achieve much better performance,

Table 3: The detailed results of Qwen3 series models on AIME, including the comparision with Speculative Decoding, where $m$ denotes the draft length. FoReaL-Decoding shows promising performance in this additional family.

| Model | | AIME24 | | | |
|---|---|---|---|---|---|
| Method | Config | ACC (%) | Length | Ratio | TFLOPs |
| **Base Models** | | | | | |
| Qwen3-32B | – | 76.6 | 13 275 | – | 15.75 |
| Qwen3-1.7B | – | 40.0 | 14 990 | – | 2.81 |
| Qwen3-0.6B | – | 13.3 | 15 839 | – | 1.14 |
| **Qwen3-32B (reasoning) + Qwen3-1.7B (reasoning)** | | | | | |
| FoReaL-Decoding | $n$=15, $p$=0.4 | 60.0 | 14 840 | 0.272 | 7.20 |
| FoReaL-Decoding | $n$=15, $p$=0.6 | 73.3 | 14 110 | 0.412 | 8.83 |
| FoReaL-Decoding | $n$=15, $p$=0.8 | 73.3 | 15 081 | 0.536 | 11.43 |
| **Qwen3-32B (reasoning) + Qwen3-0.6B (reasoning)** | | | | | |
| FoReaL-Decoding | $n$=15, $p$=0.4 | 36.7 | 17 782 | 0.281 | 7.44 |
| FoReaL-Decoding | $n$=15, $p$=0.6 | 63.0 | 14 279 | 0.410 | 8.18 |
| FoReaL-Decoding | $n$=15, $p$=0.8 | 60.0 | 15 478 | 0.560 | 11.01 |
| **Qwen3-32B (reasoning) + Qwen3-0.6B (non-reasoning)** | | | | | |
| FoReaL-Decoding | $n$=15, $p$=0.6 | 60.0 | 8 762 | 0.472 | 5.24 |
| FoReaL-Decoding | $n$=15, $p$=0.8 | 66.6 | 9 468 | 0.558 | 6.73 |
| FoReaL-Decoding | $n$=15, $p$=1.0 | 73.3 | 10 267 | 0.673 | 8.32 |
| **Speculative Decoding: Qwen3-32B (reasoning) + Qwen3-0.6B (non-reasoning)** | | | | | |
| Speculative Decoding | $m = 20$ | 73.3 | 15 374 | 0.427 | 10.25 |
| Speculative Decoding | $m = 10$ | 73.3 | 14 672 | 0.403 | 9.02 |

these results not only present the effectiveness and generalizability of FoReaL-Decoding, but also shows the potential of our method using stronger models.

## 4.6 Compared with Speculative Decoding

In this section, we compare and analyze FoReaL-Decoding with Speculative Decoding (SD) (Leviathan et al., 2023) on the Qwen3 family models. Technically, FoReaL-Decoding can be viewed as a variant of SD, but with a crucial difference in the interaction pattern between the strong and weak models. In standard SD, the weaker model first generates a fixed number of tokens $m$, after which the stronger model verifies and potentially corrects them. In contrast, FoReaL-Decoding has a stronger model that generates the initial tokens, those most likely to cause divergence, before handing off to the weaker model to complete the remainder of the sequence. As shown in the Table 3, FoReaL-Decoding achieves similar accuracy as SD but has lower estimated TFLOPs, which is mainly caused by the reduction on response lengthes, i.e., alleviate the phenomenon of overthinking. As mentioned in the motivation, SD has the same distribution with the verifier model, promising the performance but also keeping the overthinking problem.

For qualitative comparison, FoReaL-Decoding is *particularly advantageous in scenarios where misaligned tokens are **clustered together***, which is common in current reasoning models, as shown in our findings on token divergence. For instance, if there are $n$ consecutive misaligned tokens, FoReaL-Decoding requires the strong model to generate only these $n$ tokens. In contrast, SD with a draft length of $m$ would require the weak model to generate $m$ tokens, followed by verification and correction by the strong model, repeating this process until all $n$ misaligned tokens are covered. This results in $n \times m$ tokens generated by the weak model and $n$ tokens by the strong model, an $n \times m$ increase in weak model computation compared to FoReaL-Decoding.

## 5 Conclusion

Our systematic token-level analysis comparing Large Reasoning Models (LRMs) with non-reasoning models has uncovered two pivotal, previously under-explored divergence phenomena: ***Global Misalignment Rebound*** and ***Local Misalignment Diminish***. This pattern reveals a novel, periodical sentence-level pattern wherein LRM-specific stylistic "thinking cues" cause high token divergence at the very beginning of sentences, after which this misalignment rapidly decreases within the sentence. Building upon this insight, we proposed FoReaL-Decoding, a training-free, plug-and-play collaborative decoding algorithm. It allows a strong LRM to lead the crucial initial tokens of sentences (capturing these divergent "thinking cues"), while a lightweight Draft model efficiently completes the subsequent, more aligned portions. Our experiments demonstrate that FoReaL-Decoding achieves good cost-quality trade-off on reasoning-heavy math tasks.

## THE USE OF LARGE LANGUAGE MODELS

In preparing this manuscript, we utilized large language models exclusively as writing assistants to enhance clarity and presentation. The models were employed to suggest improvements to sentence structure, readability, and stylistic consistency throughout the document. Following each set of AI-generated suggestions, we carefully reviewed, evaluated, and selectively incorporated or modified the proposed changes to ensure accuracy and appropriateness. Importantly, large language models did not contribute to any aspect of research conceptualization, methodological design, experimental implementation, data analysis, or interpretation of results.

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

TABLE OF CONTENTS FOR APPENDIX

## A  PSEUDO CODE

The pseudo code of our FoReaL-Decoding is provided below, all the variables are kept the same as in the main context.

---

**Algorithm 1:** FoReaL-Decoding

---

**Input:** Leading model $P_L$, Draft model $P_D$, lead count $n$, lead probability $p$, hit threshold $k$, input prompt $q$, max new tokens MAX_LEN

**Output:** Generated tokens $\mathbf{y}$

$\mathbf{y} \leftarrow [], h \leftarrow 0, \lambda \leftarrow 0$;

$c \leftarrow q$ ;                                              // Initial context

$g \leftarrow 1$ ;                                              // Initialize gate

**while** $len(\mathbf{y}) < MAX\_LEN$ **do**

  **if** $is\_sentence\_boundary(\mathbf{y}[-1])$ **then**

    $g \sim \text{Bernoulli}(p)$ ;                   // Sample gate for new sentence

    $h \leftarrow 0$ ;                               // Reset hit counter

    $\lambda \leftarrow 0$ ;                         // Reset position in sentence

  $\lambda \leftarrow \lambda + 1$ ;                      // Increment position in sentence

  // Generate next token

  **if** $g = 1$ **and** $(\lambda \leq n$ **or** $h < k)$ **then**

    $t \leftarrow \text{sample}(P_L(\cdot|c))$ ;      // Use Leading model

  **else**

    $t \leftarrow \text{sample}(P_D(\cdot|c))$ ;      // Use Draft model

  // Check alignment when approaching transition point

  **if** $g = 1$ **and** $\lambda > n - k$ **then**

    **if** $top\text{-}1(P_D(\cdot|c)) = top\text{-}1(P_L(\cdot|c))$ **then**

      $h \leftarrow h + 1$;

    **else**

      $h \leftarrow 0$;

  $\mathbf{y}.\text{append}(t)$;

  $c \leftarrow \text{concat}(c, t)$ ;                    // Update context

  **if** $t \in EOS\_tokens$ **then**

    **break**;

**return** $\mathbf{y}$;

---

# B  FLOPs Calculation

Empirical latency depends on vendor-specific kernel fusion and memory layouts, so a timing measured on one backend may not transfer to another. Counting floating-point operations (FLOPs) provides a hardware-agnostic yardstick that isolates algorithmic differences. The performance figures we report are presented in TeraFLOPs (TFLOPs), where one TFLOP equals $10^{12}$ FLOPs. The generation process for the speculative decoding (SD)-type methods can be categories in three modes, prefill, decode, and prefix:

- *Prefill*: This stage corresponds to loading the entire prompt into the model without any KV cache. The cost here is determined by the length of the input (e.g., question and chat template) and is incurred only once per inference.

- *Decode*: This stage involves the continuous generation of new tokens, where each token is generated sequentially with the benefit of the KV cache. This is typically the most computationally intensive part of inference.

- *Prefix*: This stage occurs whenever there is a switch between models (e.g., from Drafting to Leading or vice versa), it mainly used in the SD-type methods. Here, the model must process a sequence of tokens given the KV cache of previous tokens. In SD, this happens each time the verifier model checks tokens generated by the draft model, and each time the draft model resumes generation after verification. In FoReaL-Decoding, prefix computation is triggered whenever control is transferred between the two models. Notably, the prefix stage incurs a nearly fixed cost (largely independent of the number of tokens), which is a key reason why SD-based methods can be efficient even though both models process all tokens.

The total FLOPs for a single inference is thus computed as: Prefill(Drafting) + Prefill(Leading) + Decode(Drafting) + Decode(Leading) + Prefix(Drafting $\rightarrow$ Leading) + Prefix(Leading $\rightarrow$ Drafting). When GPU memory is sufficient, profiling shows that producing multiple tokens during the prefix phase costs almost the same as decoding a single token. Therefore, we upper-bound the prefix cost by the single-token decode cost, following Speculative Thinking (Yang et al., 2025). We calculate the precise total FLOPs using the detailed formulas presented below, which is based on (Chen et al., 2024; Han, 2024; Yang et al., 2025). The resulting total FLOPs are then converted to estimated TFLOPs for reporting.

The variables involved are defined as:

- $s$: Represents the sequence length.
  - For the prefill stage ($\text{FLOPs}_{\text{prefill}}(s)$), $s$ is the length of the input prompt, denoted as $p_l$.
  - For the decode stage ($\text{FLOPs}_{\text{decode}}(s)$), $s$ is the current length of the context (prompt + tokens generated so far) that the model attends to via its Key-Value (KV) cache.
- $h$: The hidden size of the model.
- $h'$: The intermediate size of the feed-forward network (FFN).
- $n$: The number of attention heads.
- $p_l$: The length of the initial problem prompt.
- $d_l$: The number of tokens to be generated in the solution.

It is noted that the hidden size $h$ relates to the number of attention heads $n$ and the size of each attention head $d$ by $h = n \cdot d$.

The FLOPs for the prefill stage, which processes the initial input prompt of length $s = p_l$, is given by Equation 8:

$$\text{FLOPs}_{\text{prefill}}(s) = 8sh^2 + 16sh + 4s^2h + 4s^2n + 6shh' + 2sh' \tag{8}$$

The FLOPs for the decode stage, which generates a single token when the current KV cache has a length of $s$, is given by Equation 9:

$$\text{FLOPs}_{\text{decode}}(s) = 8h^2 + 16h + 4sh + 4sn + 6hh' + 2h' \tag{9}$$

The total FLOPs to generate $d_l$ tokens from a prompt of length $p_l$ combines the prefill cost for the prompt and the sum of decode costs for each generated token, as shown in Equation 10:

$$\text{FLOPs}_{\text{total}} = \text{FLOPs}_{\text{prefill}}(p_l) + \sum_{i=0}^{d_l-1} \text{FLOPs}_{\text{decode}}(p_l + i) \tag{10}$$

In this formula, for the $i$-th token being generated (0-indexed), the argument to $\text{FLOPs}_{\text{decode}}$ is $p_l + i$, representing the sequence length in the KV cache at that generation step.

## C RELATED WORKS

### C.1 LARGE REASONING MODELS

Recent advances in large language models (LLMs) have spurred a surge of work aimed at strengthening their reasoning abilities (Ahn et al., 2024; Besta et al., 2025; Chen et al., 2025a). Core reasoning skills are already instilled during pre-training, where models absorb commonsense and mathematical patterns from vast text corpora (Touvron et al., 2023; OpenAI, 2024). Researchers have therefore concentrated on post-training techniques to further polish these skills. One prominent direction employs reinforcement learning to nudge models toward more effective chains of thought (Shao et al., 2024; Xiong et al., 2025; Cui et al., 2025; Wang et al., 2025a). Another line shows that carefully curated instruction-tuning data can likewise deliver sizable gains in reasoning accuracy (Ye et al., 2025; Muennighoff et al., 2025; Wang et al., 2024a).

Despite the impressive benchmark scores of recent Reasoning Language Models, several studies have begun to probe the quality and efficiency of the reasoning they generate. (Xia et al., 2025b) conduct a broad assessment and reveal substantial redundancy in many model-produced solutions. Follow-up investigations (Chen et al., 2025c; Cuadron et al., 2025; Qu et al., 2025; Liu et al., 2025; Fan et al., 2025) underscore an "overthinking" phenomenon, whereby models craft unduly verbose derivations even for simple problems. Capitalizing on this trait, (Kumar et al., 2025) demonstrate a slowdown attack: small input perturbations can trigger excessive reasoning, markedly degrading inference speed.

To alleviate overthinking and improve efficiency for reasoning models, a series of efficient reasoning methods has been proposed. For example, (Yu et al., 2024; Team et al., 2025; Aggarwal & Welleck, 2025; Xia et al., 2025a; Luo et al., 2025) utilize model-based methods that either add further constraints on RL rewards or SFT on diverse lengths of CoTs, (Hao et al., 2024; Shen et al., 2025b;a; Zhang et al., 2025) utilize latent-space reasoning methods that transfer the massive tokens into the embedding space, (Han et al., 2024; Xu et al., 2025; Renze & Guven, 2024) utilize the prompt-based methods, (Sun et al., 2024; Wan et al., 2024; Wu et al., 2025) utilize the sampling methods. Most of these methods either require further post-training or manipulating the distribution of LRM itself.

### C.2 ALIGNMENT AND TOKEN PATTERN ANALYSIS

A key empirical foundation for LLM Alignment is LIMA (Zhou et al., 2023), which demonstrated that just $1,000$ carefully curated instruction–response pairs are already enough for LLM alignment, crystallizing the "superficial alignment" hypothesis. While a line of work directly follows the hypotheses by introducing data selection or alignment methods (Chen et al., 2023b; Li et al., 2024f; 2023; 2024b; Du et al., 2023; Li et al., 2024a; Bukharin & Zhao, 2023; Liu et al., 2023; Li et al., 2024e;c; 2025a; Xu et al., 2024), there are also works that try to further investigate this phenomenon.

(Lin et al., 2023) provides a comprehensive token-level evidence by comparing the top-k token distributions of base models and their chat-tuned counterparts. The authors show that almost all divergence concentrates on discourse markers, politeness phrases, and safety disclaimers, while core content tokens remain unchanged. (Chen et al., 2025b) dissects which prompt-level cues are sufficient (and which are not) for alignment, showing that reasoning gaps emerge precisely where superficial patterns end. The debate has sparked push-back as well: (Raghavendra et al., 2024) demonstrates systematic performance gains when the amount of post-training data scales up, arguing that some deeper representational changes do accrue beyond mere style. Researchers are also probing where superficial signals live: (Li & Kim, 2024) argues that data curation, not extra optimization steps, is the primary lever: filtering for safety disclaimers yields larger alignment jumps than adding thousands of generic examples. Together, these works paint a nuanced picture: much of the alignment gap after pre-training is indeed "superficial", residing in a narrow band of stylistic tokens that can be manipulated through tiny prompts, judicious data selection. However, in this paper, we show that *the reasoning capabilities might not be as superficial as previous findings.*

### C.3 SPECULATIVE DECODING AND COLLABORATIVE DECODING

Speculative decoding, inaugurated by (Leviathan et al., 2023), uses a small "draft" model to propose several tokens that the large "target" model then verifies in one batch, yielding $2$–$3\times$ latency

reductions with provably identical output distributions. Follow-up work, such as (Chen et al., 2023a) extends the idea to 70 B-parameter models and confirms similar speed-ups, while (Cai et al., 2024) replaces the external draft model with extra decoding heads to remove system complexity System-level schedulers like (Liu et al., 2024b) dynamically adapt draft length to traffic conditions and push end-to-end gains beyond $3\times$ in production settings.

Collaborative decoding improves text quality by letting multiple models cooperate during generation. (Li et al., 2022) runs a weak "amateur" model alongside a strong "expert" and selects tokens that maximize their likelihood gap, sharply reducing repetition and incoherence without retraining. (Jin et al., 2024) introduces a critical-token strategy that switches to the pretrained base model whenever factual precision is needed, cutting hallucinations in instruction-tuned LLMs. At an even finer grain, (Shen et al., 2024) treats "who should emit the next token" as a latent variable, enabling on-the-fly delegation between a generalist LLM and domain specialists and outperforming any single model on cross-domain tasks.

**Comparisons with related methods.** In speculative decoding (Leviathan et al., 2023), the final text provably matches what the large model alone would have produced. However, our ***FoReaL-Decoding*** focuses on the reasoning-heavy scenarios where the responses generated by the LRM itself are not desirable due to the overthinking. Thus, our method serves as a deliberate mixture of two distributions, aiming at reducing the overthinking problem of LRMs by inserting the distribution from weaker models, and at the same time increasing the efficiency. A recent work, RSD (Liao et al., 2025), also aims at reducing computation cost by utilizing speculative decoding. However, it introduces an additional process reward model as the judge, while our method focuses on utilizing the collaborative models themselves only, thus, it is largely different from our settings. Another concurrent work, Speculative Thinking (Yang et al., 2025), also shares similar motivation as ours, in which a "small-writes, large-fixes" mechanism is utilized, which differs from our "large-leads, small-follows". Moreover, ***FoReaL-Decoding*** provides a smooth transition from the small to the large model, representing wider trade-off scopes.

# D  DETAILED RESULTS

Table 4 and Table 5 show the detailed results of different settings of our method.

Table 4: The detailed results of different collaborative settings on AIME24, GPQA-D, MATH500, and AMC23, including length and ratio.

| Model | | AIME24 | | | | GPQA-D | | | | MATH500 | | | | AMC23 | | | |
|---|---|---|---|---|---|---|---|---|---|---|---|---|---|---|---|---|---|
| Method | Config | ACC (%) | Length | Ratio | TFLOPs | ACC (%) | Length | Ratio | TFLOPs | ACC (%) | Length | Ratio | TFLOPs | ACC (%) | Length | Ratio | TFLOPs |
| **DeepSeek-R1-Distill-Qwen-32B + DeepSeek-R1-Distill-Qwen-1.5B** | | | | | | | | | | | | | | | | | |
| DeepSeek-R1-Distill-Qwen-32B | | 66.7 | 13035 | - | 15.72 | 59.6 | 6602 | - | 8.09 | 93.6 | 3542 | - | 4.13 | 95.0 | 6243 | - | 7.54 |
| DeepSeek-R1-Distill-Qwen-1.5B | | 23.3 | 18021 | - | 2.86 | 22.2 | 8696 | - | 1.13 | 81.4 | 6704 | - | 1.14 | 65.0 | 13311 | - | 2.51 |
| FoReaL-Decoding | $n=15, p=0.4$ | 33.3 | 11876 | 0.272 | 5.60 | 43.3 | 5841 | 0.294 | 2.47 | 90.2 | 3402 | 0.312 | 1.45 | 80.0 | 6043 | 0.304 | 2.91 |
| FoReaL-Decoding | $n=15, p=0.6$ | 50.0 | 10934 | 0.401 | 6.77 | 48.2 | 7007 | 0.431 | 4.50 | 91.4 | 3995 | 0.452 | 2.40 | 80.0 | 6460 | 0.429 | 3.99 |
| FoReaL-Decoding | $n=15, p=0.8$ | 50.0 | 11532 | 0.527 | 8.47 | 54.6 | 6110 | 0.570 | 4.69 | 93.4 | 3658 | 0.590 | 2.70 | 90.0 | 7037 | 0.571 | 5.37 |
| FoReaL-Decoding | $n=15, p=1.0$ | 66.7 | 10617 | 0.666 | 9.16 | 56.6 | 6796 | 0.692 | 6.21 | 93.2 | 3655 | 0.726 | 3.14 | 92.5 | 5942 | 0.708 | 5.28 |
| FoReaL-Decoding | $n=25, p=0.8$ | 53.3 | 12081 | 0.676 | 10.95 | 57.7 | 6223 | 0.702 | 5.65 | 92.6 | 3585 | 0.719 | 3.13 | 92.5 | 5529 | 0.710 | 4.99 |
| FoReaL-Decoding | $n=25, p=1.0$ | 66.7 | 11116 | 0.683 | 10.54 | 57.6 | 6065 | 0.882 | 6.68 | 94.5 | 3403 | 0.890 | 3.50 | 95.0 | 5422 | 0.872 | 5.66 |
| **DeepSeek-R1-Distill-Qwen-32B + Qwen2.5-1.5B-Instruct** | | | | | | | | | | | | | | | | | |
| DeepSeek-R1-Distill-Qwen-32B | | 66.7 | 13035 | - | 15.72 | 59.6 | 6602 | - | 8.09 | 93.6 | 3542 | - | 4.13 | 95.0 | 6243 | - | 7.54 |
| Qwen2.5-1.5B-Instruct | | 0.0 | 998 | - | 0.12 | 23.7 | 923 | - | 0.12 | 49.2 | 747 | - | 0.09 | 15.0 | 818 | - | 0.10 |
| FoReaL-Decoding | $n=15, p=0.8$ | 20.0 | 12584 | 0.571 | 9.05 | 47.5 | 7013 | 0.587 | 5.63 | 76.2 | 3792 | 0.614 | 2.85 | 65.0 | 7629 | 0.514 | 5.22 |
| FoReaL-Decoding | $n=15, p=1.0$ | 20.0 | 14188 | 0.588 | 11.19 | 47.5 | 6294 | 0.737 | 5.86 | 85.9 | 3894 | 0.750 | 3.28 | 65.0 | 7673 | 0.707 | 6.15 |
| FoReaL-Decoding | $n=25, p=0.8$ | 36.7 | 11575 | 0.710 | 9.58 | 56.7 | 4718 | 0.719 | 4.37 | 82.0 | 3025 | 0.729 | 2.52 | 72.5 | 5415 | 0.649 | 4.65 |
| FoReaL-Decoding | $n=25, p=1.0$ | 40.0 | 11239 | 0.813 | 11.00 | 57.1 | 5944 | 0.887 | 6.27 | 90.8 | 3403 | 0.894 | 3.36 | 92.5 | 6989 | 0.867 | 6.88 |
| **DeepSeek-R1-Distill-Qwen-1.5B + Qwen2.5-7B-Instruct** | | | | | | | | | | | | | | | | | |
| DeepSeek-R1-Distill-Qwen-1.5B | | 23.3 | 18021 | - | 2.86 | 22.2 | 8696 | - | 1.13 | 81.4 | 6704 | - | 1.14 | 65.0 | 13311 | - | 2.51 |
| Qwen2.5-7B-Instruct | | 6.7 | 1243 | - | 0.95 | 38.4 | 1054 | - | 0.89 | 76.0 | 773 | - | 0.61 | 52.5 | 994 | - | 0.75 |
| FoReaL-Decoding | $n=15, p=0.8$ | 16.7 | 4120 | 0.545 | 2.05 | 34.3 | 2130 | 0.602 | 1.07 | 76.4 | 1341 | 0.634 | 0.57 | 57.5 | 2515 | 0.580 | 1.08 |
| FoReaL-Decoding | $n=15, p=1.0$ | 16.7 | 14132 | 0.651 | 6.47 | 29.8 | 7913 | 0.703 | 3.08 | 79.6 | 3480 | 0.735 | 1.42 | 52.5 | 7330 | 0.686 | 3.35 |
| FoReaL-Decoding | $n=25, p=0.8$ | 20.0 | 4474 | 0.693 | 1.57 | 33.1 | 1801 | 0.718 | 0.80 | 78.6 | 1498 | 0.736 | 0.55 | 65.0 | 3778 | 0.683 | 1.76 |
| FoReaL-Decoding | $n=25, p=1.0$ | 23.3 | 11436 | 0.841 | 3.18 | 29.3 | 6800 | 0.863 | 2.53 | 79.2 | 3586 | 0.891 | 1.04 | 60.0 | 5721 | 0.865 | 1.66 |

Table 5: The detailed results of different collaborative settings on AIME24 and AMC23, including length and ratio.

| Model | | AIME24 | | | | AMC23 | | | |
| Method | Config | ACC (%) | Length | Ratio | TFLOPs | ACC (%) | Length | Ratio | TFLOPs |
|---|---|---|---|---|---|---|---|---|---|
| **DeepSeek-R1-Distill-Qwen-32B + DeepSeek-R1-Distill-Qwen-1.5B** | | | | | | | | | |
| DeepSeek-R1-Distill-Qwen-32B | | 66.7 | 13035 | - | 15.72 | 95.0 | 6243 | - | 7.54 |
| DeepSeek-R1-Distill-Qwen-1.5B | | 23.3 | 18021 | - | 2.86 | 65.0 | 13311 | - | 2.51 |
| FoReaL-Decoding | $n{=}5, p{=}0.2$ | 23.3 | 12926 | 0.076 | 3.47 | 77.5 | 5634 | 0.089 | 1.39 |
| FoReaL-Decoding | $n{=}5, p{=}0.4$ | 26.7 | 11590 | 0.145 | 3.92 | 80.0 | 6549 | 0.157 | 2.18 |
| FoReaL-Decoding | $n{=}5, p{=}0.6$ | 36.7 | 11560 | 0.202 | 4.81 | 80.0 | 7081 | 0.228 | 2.95 |
| FoReaL-Decoding | $n{=}5, p{=}0.8$ | 43.3 | 11907 | 0.270 | 5.82 | 82.5 | 6399 | 0.294 | 3.29 |
| FoReaL-Decoding | $n{=}5, p{=}1.0$ | 50.0 | 13750 | 0.328 | 7.82 | 85.0 | 6916 | 0.355 | 3.86 |
| FoReaL-Decoding | $n{=}15, p{=}0.2$ | 26.7 | 12457 | 0.138 | 4.03 | 70.0 | 6680 | 0.154 | 2.05 |
| FoReaL-Decoding | $n{=}15, p{=}0.4$ | 33.3 | 11876 | 0.272 | 5.60 | 80.0 | 6043 | 0.303 | 2.91 |
| FoReaL-Decoding | $n{=}15, p{=}0.6$ | 50.0 | 10934 | 0.401 | 6.77 | 80.0 | 6460 | 0.429 | 3.99 |
| FoReaL-Decoding | $n{=}15, p{=}0.8$ | 50.0 | 11532 | 0.527 | 8.47 | 90.0 | 7037 | 0.571 | 5.37 |
| FoReaL-Decoding | $n{=}15, p{=}1.0$ | 66.7 | 10617 | 0.666 | 9.16 | 92.5 | 5942 | 0.708 | 5.28 |
| FoReaL-Decoding | $n{=}25, p{=}0.2$ | 36.7 | 10805 | 0.178 | 3.88 | 77.5 | 6798 | 0.193 | 2.32 |
| FoReaL-Decoding | $n{=}25, p{=}0.4$ | 33.3 | 11428 | 0.347 | 6.30 | 80.0 | 5929 | 0.362 | 3.17 |
| FoReaL-Decoding | $n{=}25, p{=}0.6$ | 50.0 | 10816 | 0.515 | 7.71 | 90.0 | 6169 | 0.537 | 4.49 |
| FoReaL-Decoding | $n{=}25, p{=}0.8$ | 53.3 | 12081 | 0.675 | 10.95 | 92.5 | 5529 | 0.710 | 4.99 |
| FoReaL-Decoding | $n{=}25, p{=}1.0$ | 66.7 | 11117 | 0.683 | 10.54 | 95.0 | 5422 | 0.872 | 5.66 |
| FoReaL-Decoding | $n{=}\infty, p{=}0.2$ | 30.0 | 12241 | 0.204 | 4.84 | 75.0 | 6502 | 0.216 | 2.43 |
| FoReaL-Decoding | $n{=}\infty, p{=}0.4$ | 46.7 | 11906 | 0.417 | 7.37 | 85.0 | 6719 | 0.423 | 4.07 |
| FoReaL-Decoding | $n{=}\infty, p{=}0.6$ | 50.0 | 11515 | 0.605 | 9.69 | 92.5 | 5671 | 0.607 | 4.42 |
| FoReaL-Decoding | $n{=}\infty, p{=}0.8$ | 60.0 | 10538 | 0.798 | 10.83 | 92.5 | 5925 | 0.797 | 5.87 |
| FoReaL-Decoding | $n{=}\infty, p{=}1.0$ | 66.7 | 13035 | 1.000 | 15.72 | 95.0 | 6244 | 1.000 | 7.54 |
| **DeepSeek-R1-Distill-Qwen-1.5B + Qwen2.5-7B-Instruct** | | | | | | | | | |
| DeepSeek-R1-Distill-Qwen-1.5B | | 23.3 | 18021 | - | 2.86 | 65.0 | 13311 | - | 2.51 |
| Qwen2.5-7B-Instruct | | 6.7 | 1243 | - | 0.95 | 52.5 | 994 | - | 0.75 |
| FoReaL-Decoding | $n{=}5, p{=}0.2$ | 10.0 | 1047 | 0.170 | 0.73 | 50.0 | 923 | 0.179 | 0.64 |
| FoReaL-Decoding | $n{=}5, p{=}0.4$ | 10.0 | 1381 | 0.230 | 0.91 | 55.0 | 1065 | 0.244 | 0.69 |
| FoReaL-Decoding | $n{=}5, p{=}0.6$ | 13.3 | 2377 | 0.306 | 1.61 | 62.5 | 2574 | 0.302 | 1.97 |
| FoReaL-Decoding | $n{=}5, p{=}0.8$ | 13.3 | 4203 | 0.345 | 2.87 | 47.5 | 2897 | 0.373 | 1.83 |
| FoReaL-Decoding | $n{=}5, p{=}1.0$ | 16.7 | 7236 | 0.382 | 4.45 | 50.0 | 5614 | 0.428 | 3.24 |
| FoReaL-Decoding | $n{=}15, p{=}0.2$ | 3.3 | 1936 | 0.208 | 1.38 | 47.5 | 985 | 0.224 | 0.65 |
| FoReaL-Decoding | $n{=}15, p{=}0.4$ | 16.7 | 1189 | 0.339 | 0.70 | 45.0 | 1055 | 0.360 | 0.61 |
| FoReaL-Decoding | $n{=}15, p{=}0.6$ | 16.7 | 1793 | 0.455 | 0.92 | 52.5 | 1307 | 0.482 | 0.65 |
| FoReaL-Decoding | $n{=}15, p{=}0.8$ | 16.7 | 4120 | 0.545 | 2.05 | 57.5 | 2515 | 0.580 | 1.08 |
| FoReaL-Decoding | $n{=}15, p{=}1.0$ | 16.7 | 14132 | 0.651 | 6.47 | 52.5 | 7330 | 0.686 | 3.35 |
| FoReaL-Decoding | $n{=}25, p{=}0.2$ | 13.3 | 1243 | 0.249 | 0.80 | 50.0 | 958 | 0.231 | 0.62 |
| FoReaL-Decoding | $n{=}25, p{=}0.4$ | 20.0 | 1317 | 0.389 | 0.73 | 42.5 | 1077 | 0.405 | 0.59 |
| FoReaL-Decoding | $n{=}25, p{=}0.6$ | 16.7 | 1743 | 0.536 | 0.79 | 57.5 | 2047 | 0.560 | 1.14 |
| FoReaL-Decoding | $n{=}25, p{=}0.8$ | 20.0 | 4474 | 0.693 | 1.57 | 65.0 | 3778 | 0.683 | 1.76 |
| FoReaL-Decoding | $n{=}25, p{=}1.0$ | 23.3 | 11436 | 0.841 | 3.18 | 65.0 | 5721 | 0.865 | 1.66 |
| FoReaL-Decoding | $n{=}\infty, p{=}0.2$ | 13.3 | 1072 | 0.260 | 0.69 | 50.0 | 986 | 0.290 | 0.61 |
| FoReaL-Decoding | $n{=}\infty, p{=}0.4$ | 6.7 | 1276 | 0.420 | 0.68 | 42.5 | 1140 | 0.467 | 0.56 |
| FoReaL-Decoding | $n{=}\infty, p{=}0.6$ | 10.0 | 1914 | 0.614 | 0.78 | 57.5 | 1324 | 0.618 | 0.53 |
| FoReaL-Decoding | $n{=}\infty, p{=}0.8$ | 23.3 | 4244 | 0.788 | 1.26 | 65.0 | 2854 | 0.817 | 0.79 |
| FoReaL-Decoding | $n{=}\infty, p{=}1.0$ | 23.3 | 18021 | 1.000 | 2.86 | 65.0 | 13311 | 1.000 | 2.51 |

