# OpenReview forum: "Where do Reasoning Models Make a Difference? Follow the Reasoning Leader for Efficient Decoding"
_ICLR.cc/2026/Conference — ICLR 2026 Conference Withdrawn Submission_

### Official Review · Reviewer_56oq · 2025-10-23

**Soundness:** 3
**Presentation:** 3
**Contribution:** 3
**Rating:** 4
**Confidence:** 5

**Summary:**

This paper studies where reasoning-oriented LLMs actually differ from standard instruction models. The key idea is to identify token-level misalignment patterns and exploit them for efficient decoding. Specifically, it finds that reasoning cues concentrate at sentence beginnings and proposes FoReaL-Decoding, where a strong model generates initial tokens before handing off to a smaller one. Experiments on several math reasoning benchmarks show comparable accuracy with up to 50% lower computation cost.

**Strengths:**

**[S1]** The paper provides a much-needed analysis of how reasoning and non-reasoning models differ at the token level. The observations are insightful, and the proposed method is conceptually sound and novel. The reverse interpretation of Speculative Decoding is particularly interesting.

**[S2]** The paper is overall well organized, clearly written, and easy to follow despite covering both analytical and methodological aspects.

**[S3]** The work addresses an important and timely problem (i.e., reducing overthinking and improving efficiency in reasoning LLMs).

**Weaknesses:**

**[W1`] Lack of causal analysis.** The paper identifies token-level patterns but does not clearly establish why reasoning models exhibit these behaviors; the explanations remain descriptive rather than causal.

**[W2] Evaluation bias from leader-forced alignment.** Misalignment is measured relative to the leader’s greedy outputs, which may exaggerate divergence and fail to reflect natural decoding dynamics.

**[W3] Simplistic sentence segmentation.** Sentences are detected only by punctuation or newlines, so the core sentence-initial reasoning cue assumption may break in tasks with non-standard formatting or code-like outputs.

**[W4] Only math domain.** Experiments focus almost exclusively on math reasoning datasets, leaving the generality to other reasoning types (e.g., code, commonsense) unclear.

**[W5] More important efficiency metrics.** The analysis reports theoretical TFLOPs rather than real latency (i.e., throughput), making the claimed efficiency improvements less convincing for real-world deployment.

**[W6] Hyperparameter sensitivity analysis is required.**
The gating settings (n, p, and k) are somewhat arbitrary, and robustness or automatic tuning is not explored.


I quite like the paper and believe that it has the strength to be accepted. I have recommended rejection since there are several weaknesses highlighted in the review (but I think they can be easily resolved through the rebuttal). I kindly ask the author to resolve my questions during their rebuttal.

**Questions:**

See the weakness above.

---

### Official Review · Reviewer_y5zv · 2025-10-28

**Soundness:** 2
**Presentation:** 3
**Contribution:** 2
**Rating:** 2
**Confidence:** 3

**Summary:**

This paper proposes a novel reasoning decoding method, FoReaL-Decoding, aimed at accelerating the inference of reasoning models.
The method is motivated by two key observations: Global Misalignment Rebound and Local Misalignment Diminish.
Experimental results show that the proposed approach outperforms existing methods, including lossless frameworks such as Speculative Thinking.

**Strengths:**

- The paper is clear and easy to follow, with well-presented motivation and methodology.
- The proposed observations are interesting, and the resulting decoding method seems quite effective, outperforming Speculative Thinking on reasoning benchmarks.

**Weaknesses:**

**[W1] Restricted experimental scope.** Despite the interesting observations, both the analyses and experiments are conducted only on the Qwen family. This narrow scope limits the generality and broader applicability of the findings.

**[W2] Limited logical continuity.** While the paper highlights two main observations—Global Misalignment Rebound and Local Misalignment Diminish—the proposed method is mainly motivated by the latter. The connection between the observations and the final design could be more coherently justified.

**[W3] Unclear advantage over speculative decoding.** Although the method is distinct from existing speculative decoding paradigms, its advantage remains questionable. In Table 3, the comparison with Speculative Decoding appears somewhat unfair, as the draft lengths used (10 and 20) are unusually large. A fairer comparison would involve shorter draft lengths.

**Questions:**

**[Q1]** Are the proposed observations generalizable to other reasoning models (e.g., LLaMA family) or to heterogeneous setups where the leading and draft models differ (e.g., LLaMA → Qwen)?

**[Q2]** In Table 1, only TFLOPs are reported. Could the authors also provide real latency measurements on GPUs to better quantify the practical speedup?

**[Q3]** Please re-evaluate the comparison with Speculative Decoding. Under settings where FoReaL-Decoding maintains target performance, compare with shorter draft lengths (e.g., 3 or 5) to ensure fairness and clarity of the claimed advantage.

---

### Official Review · Reviewer_dd3E · 2025-10-30

**Soundness:** 3
**Presentation:** 3
**Contribution:** 2
**Rating:** 4
**Confidence:** 4

**Summary:**

This paper introduces a novel decoding framework designed to improve the efficiency of reasoning-oriented large language models. After a token-level alignment analysis between reasoning and non-reasoning models, paper highlights two phenomena:
- Global Misalignment Rebound: Reasoning models remain stylistically and distributionally divergent even as generation length increases
- Local Misalignment Diminish: Divergence spikes at the beginning of each sentence and then quickly decays.
Leveraging these findings, the authors propose a collaborative decoding strategy where a strong reasoning model leads the initial tokens of each sentence, while a smaller model completes the decoding.
Experiments are demonstrated on math-reasoning benchmarks (AIME24, GPQA-D, MATH500, AMC23).

**Strengths:**

- Conducted token-level alignment analysis is highly informative on reasoning-specific patterns.
- Proposed framework is easy to control and allows adjustment with interpretable parameters.
- Empirical results shown that the proposed decoding strategy provides significant FLOP reduction and shorter CoTs.

**Weaknesses:**

- Reasoning domain is subjected to math reasoning. Lacks demonstration on domains with  complex problems that require unique problem solving strategies such as constraint satisfaction, MDP.
- Framework is too-dependent on the existence of a significantly stronger large reasoning model.
- Interpretation of global and local misalignments are not clearly discussed.

**Questions:**

- Including different domains into evaluation
- Mechanistic-Interpretability approaches to clearly explain the identified misalignment phenomena
- Such token level-analysis and focus on divergence spikes could have been discussed alongside with the token-level entropy metrics.

---

### Official Review · Reviewer_HLkD · 2025-10-31

**Soundness:** 3
**Presentation:** 3
**Contribution:** 3
**Rating:** 4
**Confidence:** 3

**Summary:**

This paper analyzes the differences in token distributions between reasoning models and base/instruction tuned models and finds that token distributions globally diverge from that of base/instruction tuned models with more context, but that on the sentence level, most of the difference occurs at the beginning of sentences. This motivates FoReaL-Decoding where a reasoning model is only used for the first few tokens and a non-reasoning model generates the rest of the sentence. They show that this retains most of the performance of the reasoning model while substantially reducing cost.

**Strengths:**

* The paper is clear and easy to follow.
* The two phenomena of “global misalignment rebound” and “local misalignment diminish” are potentially valuable empirical insights into how LRMs behave.
* The proposed FoReaL-Decoding method is a useful decoding method which can be used instead of traditional speculative decoding for LRMs. The method is novel in how it treats beginning and end of sentences as different.
* This method can have significant impact by reducing reasoning cost.

**Weaknesses:**

* The observed phenomena are not statistically quantified which seems important to accurately judging if this is a real phenomenon.
* The approach reminds me of s1 where the phrase “wait, “ is repeatedly added the model’s context which leads to a large performance improvement. A comparison to this type of approach where a fixed phrase is added with a certain probability as the beginning of each sentence would be useful to determine how much the “Lead” model actually contributes.
* Related to the above, it is not shown if the “Lead” model when used with FoReaL-Decoding is actually mostly outputting thinking patterns such as “wait” and “perhaps” or doing something else. Even a qualitative example of what the output from FoReal-Decoding looks like would help clarify this.
* It is not clear why reasoning length (overthinking) decreases at all with this method. The stochastic binary gate between when to use the draft model vs. lead model seems to be important for this, but the paper does not seem to provide evidence for why such a gate is useful.
* Main results in Table 1 are missing error bounds.

**Questions:**

1. For Figure 2, the difference between misalignment rebounding and what happens with the instruct and base model is subtle. Can the difference be quantified as statistically significant?
2. If the Lead model is mostly generating patterns such as “wait” and “perhaps”, then how would FoReaL-Decoding compare to an approach where the lead model is replaced with the static phrase “wait, “ which would start each sentence whenever the lead model is selected?
3. Why does reasoning length decrease using FoReaL-Decoding? Perhaps an ablation of the stochastic binary gate could answer if the gate is really the reason for the reduction in length.


Minor notes:
* The term "Local Misalignment Diminish" sounds a bit weird in the paper since it is often used as a noun phrase, since you refer to is a phenomenon. Renaming this to something like "Local Misalignment Decay" will make the sentences flow better.

---

### Note · Authors · 2026-01-05

I have read and agree with the venue's withdrawal policy on behalf of myself and my co-authors.